# Variability in BVOC emissions and air quality impacts among urban trees in Montreal and Helsinki

Kaisa Rissanen<sup>1\*</sup>, Juho Aalto<sup>2</sup>, Jaana Bäck<sup>2</sup>, Heidi Hellén<sup>3</sup>, Toni Tykkä<sup>3</sup>, Alain Paquette<sup>1</sup>

<sup>1</sup>Département des Sciences Biologiques / Centre for Forest Research, Université du Québec à Montréal, Montreal, QC H3C 3P8, Canada

<sup>2</sup>Institute for Atmospheric and Earth System Research / Department of Forest Sciences, University of Helsinki, Helsinki, 00014, Finland

<sup>3</sup>Atmospheric Composition Research Unit, Finnish Meteorological Institute, Helsinki, 00101, Finland

Correspondence to: Kaisa Rissanen (kaisa.rissanen@helsinki.fi)

<sup>\*</sup>current affiliation: Institute for Atmospheric and Earth System Research / Department of Forest Sciences, University of Helsinki, Helsinki, 00014, Finland

**Abstract.** Many cities attempt to mitigate poor air quality by increasing tree canopy cover. Trees can indeed capture pollutants and reduce their dispersion, but they can also negatively impact urban air quality. For example, trees emit biogenic volatile organic compounds (BVOCs) that participate in both ozone (O<sub>3</sub>) and secondary organic aerosol (SOA) formation, yet these emissions have been little studied in urban contexts.

We sampled BVOCs from the leaves of mature urban trees using lightweight enclosures and adsorbent tubes in two cities: Montreal, Canada and Helsinki, Finland. In both cities, we targeted five common broadleaved species in parks and streets, comparing their standardised BVOC emission potentials to nonurban BVOC emission potential estimates from emission databases-. We also calculated the potential O<sub>3</sub> and SOA formation by the study species at the leaf scale and upscaled to the neighbourhood.

We found that the measured BVOC emission potentials generally deviated little from the emission database estimates, supporting the use of database estimates for urban trees. However, tree-to-tree variation in the BVOC emission potentials was large, with slight differences between park and street trees. Compared to park trees, street tree emissions were higher in Montreal (specifically isoprene and sesquiterpenoids) and lower in Helsinki (specifically green leaf volatiles). –Finally, we found that O<sub>3</sub> formation from our study species' BVOC emissions was dominated by isoprene, while SOA formation was also affected by lower monoterpenoid and sesquiterpenoid emissions. These findings highlight the importance of species selection and management strategies that protect trees from BVOC-inducing stresses.

#### **Abbreviations**


AP, Acer platanoides, L.

BP, Betula pendula, Roth.

BVOC, biogenic volatile organic compounds

GT, Gleditsia triacanthos, L.

O<sub>3</sub>, ozone

OFP, ozone forming potential

PC, Populus x canescens, Aiton

QM, Quercus marcrocarpa, Michx

QR, Quercus robur, L.

SOA, secondary organic aerosols

SOAFP, secondary organic aerosol forming potential

TC, Tilia cordata, Mill.

TE, Tilia x europaea, Hayne

UG, Ulmus glabra, Huds.

# 1 Introduction


Poor air quality is a major health risk impacting nine out of ten people globally (WHO, 2016). Local air pollution, including both particulate matter and gaseous pollutants, is estimated to cost 8.8 million lives annually (Lelieveld et al., 2020). Particularly vulnerable to air pollution, densely populated urban areas are increasingly investing in urban greenery to improve the local air quality and living environment (see, e.g., the Million Tree Initiatives in Los Angeles, Pincetl et al., (2013)). Urban trees are expected to remove particulate matter (PM<sub>10</sub> and PM<sub>2.5</sub>, i.e., particulate matter smaller than 10 and 2.5 μm, respectively) and gaseous pollutants such as ozone (O<sub>3</sub>) through dry and wet deposition or stomatal uptake (Calfapietra et al., 2016; Nowak et al., 2014). In addition, urban trees can, for example, cool air through shading and transpiration (Pataki et al., 2021; Winbourne et al., 2020), benefit the physical and mental health of urban residents (Pataki et al., 2021; Wolf et al., 2020), and provide other ecosystem services like biodiversity conservation or flood protection (O'Brien et al., 2022).

Yet, urban trees may also negatively affect local air quality (Calfapietra et al., 2013; Churkina et al., 2017; Fitzky et al., 2019). Trees emit a variety of biogenic volatile organic compounds (BVOCs), some of which carry positive health impacts (Cho et al., 2017). Many, however, are highly reactive in the atmosphere, whereby BVOCs such as isoprene, monoterpenes, and sesquiterpenes are oxidised in reactions with hydroxide (OH) and nitrate (NO<sub>3</sub>) radicals, O<sub>3</sub>, or chlorine (Cl) (Ziemann and Atkinson, 2012). These reactions form less volatile products, which, through multiphase reactions participate in the production of secondary organic aerosols (SOAs) (Ziemann and Atkinson, 2012), constituting an important fraction of the atmospheric particulate matter (PM) (Huang et al., 2014). Moreover, although BVOCs can remove O<sub>3</sub> through oxidation, BVOC oxidation products also produce O<sub>3</sub> through reactions with nitrogen oxides (NO<sub>x</sub>) (Atkinson, 2000). Due to the abundance of NO<sub>x</sub> in urban atmospheres (Delmas et al., 1997) and the decrease in the anthropogenic VOC concentrations (EMEP database), urban O<sub>3</sub> concentrations can become increasingly sensitive to changes in BVOC concentrations (Bell and Ellis, 2004). The potential impact of BVOC emissions on both SOA and O<sub>3</sub> concentrations has been estimated in large cities (Bao et al., 2024; Ghirardo

et al., 2016; Ren et al., 2017), illustrating the important contributions of BVOCs on local O<sub>3</sub> concentrations (up to 30% in Beijing, Ren et al., 2017), albeit impacting PM<sub>2.5</sub> concentrations to a lesser degree (1.3% in Beijing, Ren et al., 2017). However, in Helsinki, monoterpenes and sesquiterpenes (either biogenic and anthropogenic VOCs) significantly contributed to SOA formation, even in urban traffic environments (Hellén et al., 2012; Saarikoski et al., 2023).

In addition to the effect of BVOCs on air quality, street tree canopies may also decelerate air flow and ventilation in street canyons, thereby delaying pollutant removal at the pedestrian level (Abhijith et al., 2017; Karttunen et al., 2020). Overall, the net impact of urban trees on air quality remains unclear and scale-dependent (Maison et al., 2024; Venter et al., 2024). To estimate the net effect of urban trees on local air quality, the positive (capture of PM and gaseous pollutants) and negative effects (BVOC effects on PM and O<sub>3</sub> production and reduced ventilation) of urban trees must be considered together. Therefore, accurate estimates of BVOC emissions of urban trees are crucial. In fact, estimates of the species-specific BVOC emission potentials (BVOC emission rate normalised to a certain temperature and light intensity), measured in natural environments or laboratory conditions, have been collected in emission databases (e.g., Kesselmeier and Staudt, 1999; Karl et al., 2009; Oderbolz et al., 2013). These databases have been used in studies estimating urban BVOC emission budgets and air quality impacts (Benjamin and Winer, 1998; Donovan et al., 2005; Owen et al., 2003; Ren et al., 2017), in tools for species selection (Churkina et al., 2015; Donovan et al., 2005; Simpson and McPherson, 2011) or for quantifying urban tree services and disservices (e.g. i-Tree tools, itreetools.org). However, because these databases contain BVOC emission potentials mainly from nonurban trees, they also carry potential sources of errors.

Firstly, the emission rates and composition of BVOCs are diverse and vary between tree genera and species, yet many tree species commonly used in urban environments lack estimates in the emission databases. While the species-level emission potentials can be approximated based on genus-level estimates (Benjamin et al., 1996), the approximations sometimes introduce large errors (Dunn-Johnston et al., 2016; Noe et al., 2008). For example, in the *Quercus* genus, there are species with mainly isoprene emissions, mainly monoterpene emissions, and species that emit a mixture of both types of isoprenoids (Loreto, 2002; Steinbrecher et al., 1997). Furthermore, species-level BVOC emission potentials and blends may differ between populations (Bäck et al., 2012; Loreto, 2002; Staudt and Visnadi, 2023), which can explain some observed differences in species-level BVOC emissions between cities (Bao et al., 2023; Cui et al., 2024; Yuan et al., 2023). More species-level data are thus needed to support species selection and to model BVOC emissions and air quality (Bao et al., 2023).

Secondly, urban environmental conditions differ from natural or laboratory conditions, sometimes exposing trees to the combined effects of drought, heat, high O<sub>3</sub> concentrations, and mechanical damage (Fitzky et al., 2019; Lüttge and Buckeridge, 2020). Because protection against and mitigation of biotic and abiotic stresses are among the most important known roles of many BVOCs (Harrison et al., 2013; Loreto and Schnitzler, 2010), these urban stress factors could induce or increase the synthesis and emissions of BVOC even in species with normally no or low emissions (Fitzky et al., 2019; Ghirardo et al., 2016; Holopainen and Gershenzon, 2010). Yet, other urban environment characteristics can be beneficial to trees, including high CO<sub>2</sub> concentration and light availability in open spaces, also impacting BVOC emissions (Bao et al., 2023; Guenther,


1997). Thus, it is uncertain how well the BVOC database values originating from nonurban environments can be applied to urban trees.

Moreover, direct measurements of urban tree BVOC emissions remain rare, primarily concentrating on trees in urban green spaces such as botanical gardens or university campuses (Ghirardo et al., 2016; Khedive et al., 2017; Noe et al., 2008; Préndez et al., 2013; Wu et al., 2021; Yuan et al., 2023; Zhang et al., 2024). A few studies have also explored BVOC emissions in varying urban green spaces or over rural-urban gradients (Duan et al., 2023; Lahr et al., 2015; Papiez et al., 2009). However, trees within the built environment have received little attention (see, however, Dunn-Johnston et al., 2016), which could bias estimates of the whole-city mean urban tree BVOC emission potentials. To quantify how well the urban tree BVOC emission potentials conform to the estimates in emission databases (species- or genus-level), more direct measurements of urban tree BVOC emissions in various urban environments are necessary. Such an understanding will allow us to estimate the net effects of urban trees on air quality more accurately and select suitable species, species compositions, and tree management strategies for urban spaces.

Here, we measured the BVOC emission rates and composition of five common urban tree species in Montreal, Canada and Helsinki, Finland, comparing the measured emission rates to earlier estimates collected from emission databases and calculating their potentials for O<sub>3</sub> and SOA formation. To quantify the variability between trees of the same species and between differing urban environments, we measured trees in two typical urban growth environments—streets and parks—in both cities.

Our specific aims were as follows:






- To provide BVOC emission potentials for common urban tree species, measured directly from the shoots of mature urban trees.
- 2) To explore the tree-to-tree variability of BVOC emissions by comparing two urban environments (parks and streets).
- 3) To compare mean urban tree BVOC emission potentials to estimates from BVOC emission databases.
- To provide O<sub>3</sub> and SOA formation potentials for the measured common urban tree species, at both the leaf and neighbourhood levels.

#### 2 Materials and methods

#### 2.1 Study sites and tree selection

The study took place in two cities: in Montreal, Canada (45.508888, -73.561668) with its 1.76 million inhabitants and in Helsinki, Finland (60.192059, 24.945831) with its 0.65 million inhabitants. The cities serve as representatives of mid-sized, high-latitude cities in the northern temperate or boreal zones – a city type that has yet to receive attention in studying urban tree BVOC emissions. Both cities have a warm summer with a humid continental climate (Dfb) based on the Köppen classification (Beck et al., 2018). Montreal's monthly mean temperatures range from 21.1°C (July) to -9.7°C (January). With

an annual precipitation of 1000 mm, approximately 270 mm falls between June and August (Environment and Climate Change Canada, 1981–2010). In Helsinki, the mean temperature ranges from 18.1°C (July) to -3.1°C (January). With its annual precipitation of 653 mm, approximately 200 mm falls between June and August (Jokinen et al., 2021, 1991–2020). In both cities, we limited the study area (~2.5 km² in Montreal and ~1 km² in Helsinki) to central residential areas, with population densities exceeding 2500 inhabitants km².






In addition, in both cities, we selected five tree species that are commonly used as urban trees within the studied climate zone and were present in both parks and streets (sidewalk pits or planting strips). Based on the literature, two species per city had moderate to high BVOC emission rates, while three had low rates. In Montreal, the high-emitting species consisted of the bur oak (*Quercus macrocarpa*, Michx., QM) and the grey poplar (*Populus x canescens*, Aiton., PC), while the low-emitting species included the Norway maple (*Acer platanoides*, L., AP), the little-leaved linden (*Tilia cordata*, Mill., TC), and the honey locust (*Gleditsia triacanthos*, L., GT). These species correspond to 0.7%, 0.3%, 15.1%, 5.4% and 9.1%, respectively, of all trees in the public tree inventory (City of Montreal open data a). In Helsinki, the high-emitting species consisted of the pedunculate oak (*Q. robur*, L., QR) and the silver birch (*Betula pendula*, Roth., BP), with the low-emitting species including the Norway maple (AP), the European linden (*T. x europaea* Hayne, TE), and the Scots elm (*Ulmus glabra*, Huds., UG). These correspond to 3.5%, 8.5%, 11.0%, 22.8% and 5.7%, respectively, of trees in the public tree inventory (City of Helsinki open data).

For each species in each city, we selected three trees from three different parks and streets within the study areas (Table 1,

Supplement Fig. S1). As an exception, all available PC trees were located inside one park and on one street. All selected trees had a minimum diameter at breast height (DBH) of 9 cm (Table 1) and were visibly healthy with no dead branches or large wounds.

Table 1. The study species in each city, the number of trees successfully sampled on streets and in parks and the range of diameter at breast height (DBH) of the study trees. An n of 2 to 3 in Montreal means that, during measurement period I or II, one sampling was unsuccessful (see Sect. 2.4).

| Montreal                   |        |       |      | Helsinki |                       |      |       |     |       |
|----------------------------|--------|-------|------|----------|-----------------------|------|-------|-----|-------|
|                            | Street |       | Park | !        |                       | Stre | eet   | Par | k     |
| Species                    | n      | DBH   | n    | DBH      | Species               | n    | DBH   | n   | DBH   |
| Quercus macrocarpa (QM)    | 3      | 16–18 | 3    | 12–44    | Quercus robur (QR)    | 3    | 9–13  | 3   | 16–48 |
| Populus x canescens (PC)   | 3      | 16–18 | 3    | 15–17    | Betula pendula (BP)   | 3    | 19–26 | 3   | 21–54 |
| Acer platanoides (AP)      | 2 to 3 | 14–28 | 3    | 19–44    | Acer platanoides (AP) | 3    | 13–29 | 3   | 24–53 |
| Tilia cordata (TC)         | 3      | 32–42 | 3    | 20-42    | Tilia x europaea (TE) | 3    | 11-25 | 3   | 21–48 |
| Gleditsia triacanthos (GT) | 3      | 11–13 | 2-3  | 31–34    | Ulmus glabra (UG)     | 3    | 24–43 | 3   | 14–74 |

# 2.2 BVOC sampling and auxiliary measurements

# 2.2.1 BVOC sampling







We sampled BVOC emissions from the Montreal trees twice—on 2 to 15 June (period I) and 11 to 25 August (period II) 2022—and from Helsinki trees once on 6 to 25 July 2022 (for exact sampling dates, see Table S1). The sampling days were preferably sunny—without rain or heavy cloud cover—and sampling always took place between 11.00 and 15.00. Ambient temperatures during sampling were 19 to 33 °C in period I and 21 to 37 °C in period II in Montreal, and 22 to 33 °C in Helsinki (Supplement Table S1).

From each tree, we sampled BVOC emissions from one undamaged shoot with mature leaves in a sunny position in the lower or mid-canopy area. We carefully enclosed the shoot in a polyethylene terephthalate (PET) bag (LOOK oven bag 35 x 43 cm, heated at 120°C for at least 1 h, (Vedel-Petersen et al., 2015); for recovery tests, see Supplement mMethods S1 and Table S2), with a 6.35 mm diameter fluorinated ethylene propylene (FEP) tubing for replacement and sample air (Fig. S2). We first flushed the bag at a flow rate of 2 L min<sup>-1</sup> for 15 minutes and then sampled side streams of both the replacement and sample air into adsorbent tubes (Tenax TA and Carbopack B) with a flow rate of 0.08 L min<sup>-1</sup> for 30 minutes (keeping the flow rate through the bag at 2 L min<sup>-1</sup>). The sampling time was reduced to 15 minutes for QM and PC in Montreal in August to avoid isoprene saturation. For quality control of any impurities in the tubing and measurement system, we also sampled an empty bag at the beginning of each sampling day using the same setup. To pump the replacement air, controlling and logging the air flowrates, and collecting the air samples into adsorbent tubes, we used a custom-made gas sampler. The replacement air relied on non-filtered ambient air to maintain close-to-ambient conditions within the sampling bag and to allow for potential ozone effects on leaf BVOC emissions. To ensure BVOC stability in the adsorbent tubes during sampling and storage, we added sodium thiosulfate-impregnated filters to remove any ozone before reaching the adsorbent tube (Hellén et al., 2024). Between 7 June and 25 July, we could only use the filters for outgoing sample air adsorbent tubes because of supplier delays. We corrected this imbalance in the BVOC emission calculations (see methods Sect. 2.3).

After sampling, we stored the adsorbent tubes and the sampled branch in a box cooler until the end of the day, after which the tubes were stored in a refrigerator (~4 to 8°C). We photographed the sampled shoot leaves for leaf area calculation and dried them at 60°C for at least 48 h before weighing their dry mass.

# 2.2.2 Auxiliary measurements

During sampling, we tracked the temperature, humidity, and light conditions (photosynthetically active radiation, PAR) within and outside the sampling bag by placing a humidity/temperature sensor (Rotronic HC2-S3C03, Bassersdorf, Switzerland) within the sampling bag and a quantum sensor (LI-190R-SMV-5, LI-COR Biosciences, Lincoln, NE, USA) beside the bag. In addition, we logged the ambient air temperature and humidity outside the bag using a sensor (RuuviTag pro, Riihimäki, Finland) shaded from direct solar radiation. We measured the ambient air O<sub>3</sub> concentration during BVOC sampling using a multi-gas monitor (Gasmaster Gas Monitor 2710 with O<sub>3</sub> head from 0–0.15 ppm, Kanomax, NJ, USA). Following BVOC

sampling, we measured the leaf water potential of three leaves near the sampled branch using a portable pressure chamber (Pump-Up Chamber, PMS Instrument, OR, USA). For information on the meteorological conditions before and during the sampling periods, we accessed the temperature and precipitation records of the Université du Québec à Montréal weather station (Université du Québec à Montréal / ESCER) and the Finnish Meteorological Institute Kumpula weather station (Finnish Meteorological Institute).

# 2.3 BVOC analysis and calculations

We analysed the adsorbent tubes by using a gas-chromatograph mass-spectrometer (GC-MS; Clarus 680 and Clarus SQ T or Clarus SQ 8 C, PerkinElmer, Waltham, MA, USA) with thermal desorption (TurboMatrix 350, PerkinElmer, Waltham, MA, USA) at the Finnish Meteorological Institute within 30 days of sample collection. Relying on the analytical method of Helin et al. (2020), we used six calibration standards of 25 compounds in each GC-MS run to quantify the compound concentrations in the sample tubes (see the list of detected and calibrated compounds in Table S3) and identified the compounds by comparing their retention times and mass spectra to the calibrated standard. For isoprene, we used one gaseous calibration standard (National Physical Laboratory, Teddington, UK). Compounds which were not in the calibration standards were tentatively identified by comparing their retention times and mass spectra with the National Institute of Standards and Technology (NIST, Gaithersburg, Maryland, USA) mass spectral library, and calculated as the calibrated compound closest in composition and retention time (Table S3). When the tentative identification did not provide a confident match, we grouped the compound into a subgroup following Guenther et al. (2012) (hemiterpene, monoterpenoid, sesquiterpenoid, GLV).

Two corrections to the measured BVOC concentrations were necessary. First, as mentioned in Sect. 2.2.1, samples collected between 7 June and 25 July did not include ozone removal for the incoming replacement air samples. For these samples, we corrected the measured BVOC concentrations using terpenoid losses due to ozone reactivity in adsorbent tubes quantified by Helin et al. (2020) and the measured ambient ozone concentrations to correct the concentration values (for details, see Supplement mMethods S2). The potential error and the correction likely had a minor impact on the calculated emission rates (Fig. S3). Second, in 13 samples from QM, QR, and PC, the isoprene concentrations in the outgoing samples were higher than the isoprene standard. In nine of these samples, the concentration saturated the selected ion monitoring (mass 67), for which we thus quantified and calibrated the peak using the ion scan mode (mass 65). In these samples, the isoprene concentrations are less certain than in other samples and compounds, although they serve as the conservative minimum estimates of the isoprene concentration. In addition, we did not use the samples from two extreme cases (one PC on the street and one QR in the park) in further analyses.

In Eq. (1), we calculated the shoot emission rates ( $E_{shoot}$ ) or empty bag emission rates ( $E_{bag}$ ) for each compound based on the difference in mass of the incoming replacement air and outgoing sample ( $C_{in}$  and  $C_{out}$  ng), the sampling air flowrate ( $F_{s}$  L min<sup>-1</sup>), the sampling time (t, min), and the replacement air flowrate ( $F_{R}$ , L min<sup>-1</sup>) as follows:.

$$E_{\text{shoot or bag}} = \left(\frac{C_{\text{out}}}{F_{\text{S}}^*t} - \frac{C_{\text{in}}}{F_{\text{S}}^*t}\right) * F_{\text{R}}$$
(1).

To account for impurities or the retention of compounds within the sampling system, we subtracted the empty bag emission rates from the shoot emission rates measured on the same day ( $E_{shoot\ corrected} = E_{shoot} - E_{bag}$ ). Finally, we divided the corrected shoot emission rates by the dry leaf mass and multiplied it by 60 to obtain the emissions per dry mass in an hour (ng g<sup>-1</sup> DW h<sup>-1</sup>).

We calculated the detection limits of the sampling system as 3x the standard deviation (SD) of all incoming sample concentrations masses (Table S3). When the concentration mass difference between the incoming and outgoing samples (Cout - Cin) did not exceed the detection limit, we flagged the compound emissions as uncertain. We included the flagged compounds to calculate the total emission rates over the compound groups to avoid a consistent negative bias, but did not use them in compound-specific analyses. We also calculated the analytical detection limits per measurement period across all the TD-GC-MS runs during the period as the mean peak concentration of empty tube measurements + 3x SD for analytical quality control (Table S3).

# 2.4 Data analysis




To prepare the shoot emission data for further analysis, we first removed any data with potential signs of rough handling in the BVOC emissions because of their sensitivity to mechanical stress. To do so, we plotted the emission rates (isoprene, monoterpenoid total, sesquiterpenoid total and green leaf volatile (GLV) total) of all trees per species against the temperature and PAR while removing trees which had abnormally high emissions (3- to 100-fold emission rates) for the given temperature or PAR. This removed one GLTR in a park in Montreal sampled during period I and one ACPL measured on a street in Montreal during period II. Isoprene, monoterpenoid and GLV emission rates of both trees were also identified as influential points by Bonferroni outlier test (outlierTest of R package car, version 3.1.1) for a linear model against temperature and PAR corrections (G97 functions, Guenther, 1997). Next, we calculated the emission potentials, that is, the emission rates normalised to temperature 30°C and PAR 1000 μmol m<sup>-2</sup> s<sup>-1</sup> based on G97 functions (Guenther, 1997). The occurrence of low light conditions (PAR< 50 μmol m<sup>-2</sup> s<sup>-1</sup>) during sampling can inflate uncertainty in the normalisation. Thus, we removed one ACPL and one TICO measured on streets in Montreal during period I and one ACPL measured on streets in Montreal during period II from the analysis that used the emission potentials.

# 2.4.1 Differences between park and street tree BVOC emission potentials and BVOC emission potential correlations with environmental factors

To explore the effect of the site type (park or street) on the emission potentials of isoprene, and the monoterpenoid, sesquiterpenoid, and GLV totals, we used the analysis of variance (ANOVA) F-test (Anova of R package car, version 3.1.1).

We applied the test first separately by city and in Montreal by measurement period (I or II), using site type, species, and the interaction between species and site type as explanatory variables ( $n_{obs}$ = 26 to 30). We applied log + i-transformation (where i was a small number, 0.5x the smallest nonzero value) on the emission potential data for a normal distribution of the residuals (calculated using Shapiro–Wilk test, Shapiro.test of R package stats, version 4.2.2).

In addition, to explore the potential small-scale impacts of the tree environment and potential stress sources across the two site types, we calculated Pearson's correlations between environmental factors around each tree and the emission potentials of isoprene, and the monoterpenoid, sesquiterpenoid, and GLV totals, separately per city, measurement period and species. The environmental factors included the ambient temperature, PAR and O<sub>3</sub> concentration during sampling, as well as the degree of impermeability around the tree. For Helsinki, we calculated the degree of impermeability for a 10-m radius around the tree stem using (Helsinki Region Environmental Services open data), and in Montreal, we used the mean value per city block (City of Montreal open data b). Finally, we also correlated the BVOC emission potentials against leaf water potential (WP).

# 2.4.2 Differences between measured urban tree BVOC emission potentials and estimates from databases

We collected the estimates of the species- or genus-level emission potentials for isoprene, and the monoterpenoid and sesquiterpenoid totals from several BVOC emission databases (Table S4). We then calculated the mean for all unique database values per species and compound group for the "reference emission potential" (see Tables 3 and 4), which we compared with our measured urban tree BVOC emission potentials. In this comparison, we used one-sample Wilcoxon tests (wilcox.test function of R-package stats, version 4.2.2), separately per city, species and compound group, and the sampling period in Montreal (nobs = 4 to 6 per species). Although the Wilcoxon test accommodates small sample sizes and nonnormal distributions of the emission data, the results here should be interpreted with care.

# 2.4.3 Potential air quality effects of the study species





To estimate the  $O_3$ -formation potential per tree and compound, we used OFP = E x MIR, where OFP is the  $O_3$ -formation potential ( $\mu$ g g<sup>-1</sup> h<sup>-1</sup>), E is the emission potential of the BVOC ( $\mu$ g g<sup>-1</sup> h<sup>-1</sup>), and MIR is the maximum incremental reactivity (g g<sup>-1</sup>), which is based on previous work by Carter, (1994, 2010). OFP estimates the maximum potential of BVOC to contribute to the production of photochemical  $O_3$  (Carter, 1994, 2010). When a compound-wise MIR was not available for a monoterpenoid in our study, we used the average value of 4.04 g g<sup>-1</sup> (Carter, 1994, 2010). Following Su et al. (2016), Wang et al. (2019) and Yang et al. (2023), for sesquiterpenoids, we used the value of 1.71 g g<sup>-1</sup> for C15 alkenes. To estimate the SOA-formation potential (SOAFP) per tree and compound, we used SOAFP = E x FAC, where SOAFP is the SOA-formation potential, E is the emission potential of BVOC ( $\mu$ g g<sup>-1</sup> h<sup>-1</sup>), and FAC is the fractional aerosol coefficient (%, the fraction of BVOC which converts into aerosol, according to Grosjean (1992) and Grosjean and Seinfeld (1989)). We collected the compound-wise FAC values from Grosjean (1992), Hoffmann et al. (1997), Griffin et al. (1999), and Carlton et al. (2009). When a compound-wise FAC was unavailable, we used the mean over the compound group (monoterpenoids or

sesquiterpenoids). When we found multiple FAC estimates per compound, we used the mean across estimates. The MIR and FAC values we used per compound are listed in Table S5.

We summed the compound-wise OFP and SOAPF per tree to present the total OFP and SOAFP. We then applied the ANOVA F-test to explore the differences in OFP and SOAFP between species and site types ( $n_{obs}$ = 26 to 29), similar to what we did for the BVOC emission potentials (see Sect. 2.4.1).

# 2.4.4 Upscaling BVOC emission potentials, OFP and SOAFP


310

290 Finally, to explore the contributions of the study species when accounting for their presence in the study areas, we calculated rough estimates for the total BVOC emission, OFP, and SOAFP per land area for the study species in Montreal and Helsinki. These estimates do not serve to approximate the real BVOC emissions of urban tree populations or their OFP and SOAFP, because they only include our study species and tree individuals present in the public tree databases, and because of the simplified presentation of the tree canopies, but they allow for tentative comparisons between species. We first selected 295 smaller, representative areas of interest ("upscaling test areas") within the study areas: 1.75 km<sup>2</sup> in Montreal and 0.30 km<sup>2</sup> in Helsinki (Fig. S1). We selected the upscaling test areas to consist mainly of street or park land uses (excluding most green areas with trees not listed in the public tree databases) and to contain individuals from our study species. We then approximated the canopy area of each tree within the area of interest by cutting the canopy cover layer (estimated based on an aerial image and aerial LiDAR data, in Montreal from 2019 (City of Montreal open data c), and in Helsinki combining estimates from 2020 300 and 2022 (Helsinki Region Environmental Services open data)) using Voronoi polygons defined by tree location layer from public tree databases in OGIS (OGIS Development Team 2025, version 3.40.5). To minimise the inclusion of tree canopies not listed in the public tree databases, we limited the maximum canopy radius to 9 m. Using a leaf area index of 5 m<sup>2</sup> m<sup>-2</sup> (as used in the MEGAN biogenic emission model, Guenther et al. (2012)), we then converted the tree-wise canopy areas to treewise leaf areas. We used the tree-wise leaf area (m<sup>2</sup>), species-wise leaf mass per area (calculated from a measured leaf area and mass, g m<sup>-2</sup>), and the measured species-wise BVOC emission potential, OFP, or SOAFP (µg g<sup>-1</sup> h<sup>-1</sup>) to calculate the total 305 tree-wise emission potentials of isoprene, monoterpenoids, and sesquiterpenoids along with OFP and SOAFP for each tree in our study species.

We corrected the tree-wise emission potentials by a factor of 0.75 assuming that half of the tree leaf area is shaded at any given time of day and thus emits at a rate that is half the measured emission potential of the sun leaves [1 - (0.5 \* 0.5) = 0.75]. In addition, we calculated the tree-wise emission potentials without correction (assuming all of a tree leaf area emits at the rate of the measured emission potential of the sun leaves) and at a higher factor of 0.5 (assuming half of the tree canopy is shaded and does not emit BVOC). Finally, we summed the corrected emission potentials of all trees from our study species and divided the sum by the area of the upscaling test area to determine the BVOC emission, and the OFP and SOAFP intensities of the study species as  $\mu g m^{-2}$  (land area)  $h^{-1}$ .

# 315 3 Results

320

325

330

#### 3.1 Environmental conditions before and during BVOC sampling

Monthly mean temperatures in June (19.2°C) and August (21.8°C) 2022 in Montreal were slightly higher than the long-term mean monthly temperatures (18.6°C and 20.1°C, respectively, Figs. S4a–b). The ambient and chamber temperatures tended to be higher during the street tree than park tree sampling (Fig. S5a). In Helsinki, the July 2022 mean temperature (18.5°C) approximated the long-term mean (18.1°C), and the ambient or chamber temperatures did not differ between the park and street trees (Figs. S4c and S5a).

The precipitation totals in June (165 mm) and August (104 mm) in Montreal slightly exceeded the long-term monthly totals (87 mm and 94 mm, respectively, Figs. S4a–b). In June, however, most precipitation fell after the sampling period. In Helsinki, the July precipitation total (49 mm) was slightly lower than the long-term average (57 mm, Fig. S4c). With close-to-average precipitation, trees in both cities were unlikely to experience drought. Correspondingly, mid-day leaf water potentials remained mostly above -2 MPa (Fig. S5b). The lowest water potentials (-2.1 MPa) occurred in Montreal in August (Fig. S5b). Ambient O<sub>3</sub> concentrations during sampling varied between 0.010 and 0.042 ppm in Montreal and between 0.019 and 0.039 ppm in Helsinki (Fig. S5a), thus remaining mostly below harmful levels compared with the European Environment Agency AOT40 standard threshold of 0.04 ppm and Canadian Ambient Air Quality Standards threshold of 0.06 ppm. For comparison, daytime ambient O<sub>3</sub> concentrations measured from rural background air near Helsinki (Luukki, Espoo) were on average 0.034 ppm in summer 2022 (https://en.ilmatieteenlaitos.fi/download-observations). In Montreal, ambient O<sub>3</sub> concentrations were

#### 3.2 BVOC emission rates in urban trees

#### 3.2.1 Measured BVOC emission potentials and variation between park and street trees

The total BVOC and isoprene emission potentials were high (>20 ng g<sup>-1</sup> h<sup>-1</sup>) for PC and QM in Montreal (Figs. 1a–d) and for QR in Helsinki (Figs. 2a–b), but low for other species. The isoprene emission potentials were negligible (<0.1 ng g<sup>-1</sup> h<sup>-1</sup>) for AP, GT, TC, TE, and UG (Figs. 1c–d<sub>7</sub> and Fig. 2b). The monoterpenoid emission potentials were highest for BP in Helsinki (Fig. 2c) and low (<1 ng g<sup>-1</sup> h<sup>-1</sup>) for the other species (Figs. 1e–f and Fig. 2c). The sesquiterpenoid emission potentials were generally low (<0.5 ng g<sup>-1</sup> h<sup>-1</sup>), except for TC in Montreal and AP in Helsinki (Figs.1g–h and 2d), which were of a similar magnitude or even higher than the emissions of monoterpenoids. The GLV emission potentials were highest among AP in both cities, but with large tree-to-tree and temporal variation (Figs. 1i–j and 2e).

generally higher during street tree than park tree sampling, while in Helsinki we observed no difference (Fig. S5a).

Figure 1. BVOC emission potentials across all compound groups (a-b) and separately for isoprene (c-d), monoterpenoids (e-f), sesquiterpenoids (g-h) and green leaf volatiles (GLVs, i-j), for study species (QM, Quercus marcrocarpa; PC, Populus x canescens; TC, Tilia cordata; AP, Acer platanoides; GT, Gleditsia triacanthos) in parks and streets in Montreal, measured in June (period I: a, c, e, g, and i) and August (period II: b, d, f, h, and j) 2022. The error bars show the 95% confidence intervals for the mean emission potentials. For each bar, nobs = 2-3.

Figure 2. BVOC emission potentials across all compound groups (a) and separately for isoprene (b), monoterpenoids (c), sesquiterpenoids (d), and green leaf volatiles (GLVs, e) for the study species (QR, *Quercus robur*; BP, *Betula pendula*; AP, *Acer platanoides*; UG, *Ulmus glabra*; TE, *Tilia x europaea*) in parks and streets in Helsinki, in July 2022. The error bars show the 95% confidence intervals for the mean emission potentials. For each bar, nobs = 3.

The total BVOC emission potentials appeared generally higher for street than park trees in Montreal, except for QM (Fig. 1), but higher for park than street trees in Helsinki, except for UG (Fig. 2). Across all tree species, the isoprene and sesquiterpenoid emission potentials were higher for street than park trees in Montreal during period II ( $F_{1,22} = 5.91$ , p = 0.024 for isoprene;  $F_{1,23} = 5.00$ , p = 0.042 for sesquiterpenoids, Fig. 1h, Table 2), specifically for PC (isoprene, Fig. 1d) and TC (sesquiterpenoids, Fig. 1h). In Helsinki, the GLV emission potentials were higher for park than street trees ( $F_{1,24} = 6.72$ , p = 0.016, Table 2), specifically for AP and BP (Fig. 2e). The monoterpenoid emission potential differences between park and street trees were varied and depended upon the species, particularly in Helsinki (for site type and species interaction,  $F_{4,20} = 6.72$ , p = 0.016, Table 2, Fig. 2c).

Table 2. Analysis of variance (ANOVA) F-tests for the effects of the site type, species, and their interaction on the BVOC emission potentials of isoprene and for the monoterpenoid, sesquiterpenoid, and green leaf volatile (GLV) totals<u>sums</u>, and ozone-formation potential (OFP) and SOA-formation potential (SOAFP) per city and month. SS, Sum of Squares; DF, degrees of freedom; F, F-value; p, p-value. The interaction term was removed if it was not significant in the ANOVA type III test (p > 0.05). Significant effects (p < 0.05) are highlighted in bold.

| ANOVA F-tests    |                       | Mont | real Pe      | riod I |            | Monti | real Pe      | riod II |            | Helsi | nki          |      |            |
|------------------|-----------------------|------|--------------|--------|------------|-------|--------------|---------|------------|-------|--------------|------|------------|
|                  |                       | SS   | D <u>F</u> f | F      | <u>p</u> P | SS    | D <u>F</u> ₽ | F       | <u>p</u> P | SS    | D <u>F</u> f | F    | <u>p</u> P |
| Isoprene         | Site type             | 0.12 | 1            | 0.12   | 0.732      | 5.47  | 1            | 5.91    | 0.024      | 1.55  | 1            | 0.56 | 0.462      |
|                  | Species               | 403  | 4            | 97     | < 0.001    | 493   | 4            | 133     | <0.001     | 334   | 4            | 30   | < 0.001    |
|                  | Residuals             | 20.8 | 20           |        |            | 20.4  | 22           |         |            | 63.7  | 23           |      |            |
| Monoterpenoids   | Site type             | 0.29 | 1            | 0.62   | 0.441      | 1.82  | 1            | 1.22    | 0.281      | 0.33  | 1            | 0.44 | 0.513      |
| •                | Species               | 18.0 | 4            | 9.65   | < 0.001    | 17.8  | 4            | 2.98    | 0.042      | 53.5  | 4            | 18.1 | <0.001     |
|                  | Site type:<br>Species |      |              |        |            |       |              |         |            | 12.1  | 4            | 4.11 | 0.014      |
|                  | Residuals             | 9.81 | 21           |        |            | 32.8  | 22           |         |            | 14.8  | 20           |      |            |
| Sesquiterpenoids | Site type             | 0.90 | 1            | 0.65   | 0.428      | 8.68  | 1            | 5.00    | 0.036      | 0.69  | 1            | 0.30 | 0.586      |
|                  | Species               | 16   | 4            | 2.96   | 0.044      | 6.02  | 4            | 0.87    | 0.500      | 5.77  | 4            | 0.64 | 0.642      |
|                  | Residuals             | 29.0 | 21           |        |            | 38.2  | 22           |         |            | 54.5  | 24           |      |            |
| GLVs             | Site type             | 0.02 | 1            | 0.02   | 0.902      | 1.58  | 1            | 1.90    | 0.182      | 7.72  | 1            | 6.72 | 0.016      |
|                  | Species               | 7.44 | 4            | 1.59   | 0.215      | 3.83  | 4            | 1.15    | 0.358      | 19    | 4            | 4.23 | 0.010      |
|                  | Residuals             | 24.6 | 21           |        |            | 18.3  | 22           |         |            | 27.6  | 24           |      |            |
| OFP              | Site type             | 0.07 | 1            | 0.10   | 0.754      | 5.30  | 1            | 7.74    | 0.011      | 0.18  | 1            | 0.19 | 0.668      |
|                  | Species               | 188  | 4            | 64     | < 0.001    | 321   | 4            | 117     | < 0.001    | 196   | 4            | 53   | < 0.001    |
|                  | Residuals             | 14.7 | 20           |        |            | 15.1  | 22           |         |            | 21.5  | 23           |      |            |
| SOAFP            | Site type             | 0.53 | 1            | 0.57   | 0.459      | 4.59  | 1            | 8.41    | 0.008      | 0.33  | 1            | 0.16 | 0.690      |
|                  | Species               | 36   | 4            | 9.72   | < 0.001    | 114   | 4            | 52      | <0.001     | 53.7  | 4            | 6.64 | 0.001      |
|                  | Residuals             | 18.6 | 20           |        |            | 12.0  | 22           |         |            | 46.5  | 23           |      |            |

Within site type and species, tree-to-tree variation in BVOC emission potentials was often large (Figs. 1 and 2). Explaining some of the variation, BVOC emission potentials showed species-specific correlations with local environmental factors and leaf water potential, used to characterise potential sources of stress (Fig. 3). Ambient temperatures or PAR often correlated positively with terpenoid emission potentials in Montreal, in particular for TC and AP in Period I (Figs. 3c–d) and PC and TC in Period II (Figs. 3g–h). Despite the generally low O<sub>3</sub> concentrations, they correlated positively with the sesquiterpenoid emission potentials of AP in Montreal both in Period I and II (in period II, marginally significant) (Figs. 3d and 3i). The degree of impermeability correlated positively with PC isoprene emission potentials in Montreal in Period II (Fig. 3g), but interestingly negatively (although marginally) with sesquiterpenoid and GLV emission potentials of BP in Helsinki (Fig. 3l). Although we did not detect signs of drought, leaf water potential correlated positively with monoterpene emission potentials among QM in Montreal in Period II (Fig. 3f).

Figure 3. Pearson's correlation coefficients between the emissions potentials of compound groups and environmental (Tamb, ambient temperature during sampling; PAR, photosynthetically active radiation during sampling; O<sub>3</sub>, ambient ozone concentration during sampling; % imperm, percentage of surface impermeability within a 10-m radius around the tree (Helsinki) or mean surface impermeability of the city block (Montreal)) and physiological (WP, shoot water potential) factors as representatives of sources or indicators of stress. Correlations were calculated separately by species, city and measurement period. Species: AP, Acer platanoides; GT, Gleditsia triacanthos; PC, Populus x canescens; QM, Quercus macreerocarpa; TC, Tilia cordata; BP, Betula pendula; QR, Quercus robur; TE, Tilia x europaea; UG, Ulmus glabra.

# 3.2.2 Comparison to emission databases

The isoprene, monoterpenoid, and sesquiterpenoid emission potentials of urban trees lay primarily near or below the reference emission potentials (Fig. 4, Tables 3 and 4). More specifically, the isoprene emission potentials of QM and PC in Montreal during period I (Fig. 4a), the monoterpenoid emission potential of AP in both cities (Figs. 4d–f) and the sesquiterpenoid emission potential of BP in Helsinki (Fig. 4i) were lower than expected based on the database values. Among PC, the low isoprene emission potentials were mainly driven by park trees (Figs. 4a–b). The largest positive deviations from the reference emission potentials were for the QM isoprene emission potentials in Montreal during period II (Fig. 4b) and the QR isoprene emission potentials and the BP monoterpenoid emission potentials, mostly because of park trees, in Helsinki (Figs. 4c and 4f). In addition, the sesquiterpenoid emission potentials were slightly higher than the reference emission potentials for TC in Montreal during period I, mainly caused by street trees (Fig. 4g), and for AP in Helsinki (Fig. 4i). Given the large tree-to-tree variation, these differences were, however, not significant (Tables 3 and 4). Significant but small positive deviations from the reference emission potentials were found for the PC and TC monoterpenoid emission potentials in Montreal, mostly driven by street trees (Figs. 4d–e, Table 3).

Figure 4. The difference between the isoprene (a-c), monoterpenoid (d-f), and sesquiterpenoid (g-i) emission potentials measured from urban trees and the reference values (mean value of the emission potentials collected from the emissions databases), separately for species in Montreal in June (period I: a, d and g) and August (period II: b, e, and h) and in Helsinki in July (c, f, and i). The filled

dots indicate the mean difference across the trees (nobs = 4-6) and the error bars show the 95% confidence intervals (some are smaller than the dot diameter). The open triangles show the difference for each park tree and the open dots for each street tree. Species: AP, Acer platanoides; GT, Gleditsia triacanthos; PC, Populus x canescens; QM, Quercus marcrocarpa; TC, Tilia cordata; BP, Betula pendula; QR, Quercus robur; TE, Tilia x europaea; UG, Ulmus glabra.

Table 3. The emission potentials of isoprene and monoterpenoid and, sesquiterpenoid sums as the mean (standard deviation) across the trees measured per species and measurement period (Period I in June and Period II in August) in Montreal in 2022, as  $\mu g$   $g(DW)^{-1}$   $h^{-1}$ , at T 30°C and PAR 1000  $\mu$ mol m<sup>-2</sup> s<sup>-1</sup>. In the Wilcoxon test, the measured values were compared against the reference value (mean value across the values listed in Table S4); significant deviations (p < 0.05) are highlighted in bold. AP, *Acer platanoides*, GT, *Gleditsia triacanthos*; PC, *Populus x canescens*; QM, *Quercus marcrocarpa*; TC, *Tilia cordata*. Nobs number of observations (trees) in the Wilcoxon test.

|                  | Species | Measured n<br>emission por |             | Reference<br>BVOC     | Wilcoxon test p-values (n <sub>obs</sub> ) |           |  |  |
|------------------|---------|----------------------------|-------------|-----------------------|--------------------------------------------|-----------|--|--|
| Compound         |         | Period I                   | Period II   | emission<br>potential | Period I                                   | Period II |  |  |
| Isoprene         | AP      | 0.01 (0.00)                | 0.02 (0.01) | 0.07                  | 0.063 (5)                                  | 0.125 (4) |  |  |
| -                | GT      | 0.02 (0.01)                | 0.05 (0.07) | 0.11                  | 0.063 (5)                                  | 0.188 (6) |  |  |
|                  | PC      | 52.2 (20.7)                | 65.6 (42.5) | 74.7                  | 0.063 (5)                                  | 0.843 (6) |  |  |
|                  | QM      | 30.1 (19.9)                | 93.3 (38.7) | 74.2                  | 0.031(6)                                   | 0.313 (6) |  |  |
|                  | TC      | 0.01 (0.01)                | 0.01 (0.02) | 2.75                  | 0.125 (5)                                  | 0.125 (6) |  |  |
| monoterpenoids   | AP      | 0.13 (0.11)                | 0.07 (0.06) | 1.72                  | 0.063 (5)                                  | 0.125 (4) |  |  |
|                  | GT      | 0.45 (0.29)                | 0.04 (0.03) | 0.46                  | 1 (5)                                      | 0.031 (6) |  |  |
|                  | PC      | 0.48 (0.21)                | 0.27(0.27)  | 0.07                  | 0.031(6)                                   | 0.156(6)  |  |  |
|                  | QM      | 0.06 (0.03)                | 0.12 (0.16) | 0.19                  | 0.031(6)                                   | 0.438 (6) |  |  |
|                  | TC      | 0.48 (0.57)                | 0.83 (1.42) | 0                     | 0.063 (5)                                  | 0.031 (6) |  |  |
| sesquiterpenoids | AP      | 0.14 (0.12)                | 0.08 (0.07) | 0.1                   | 0.625 (5)                                  | 0.625 (4) |  |  |
|                  | GT      | 0.08 (0.07)                | 0.02 (0.01) | 0.03                  | 0.188 (5)                                  | 0.156 (6) |  |  |
|                  | PC      | 0.11 (0.07)                | 0.05 (0.06) | 0.1                   | 1 (6)                                      | 0.094(6)  |  |  |
|                  | QM      | 0.03 (0.03)                | 0.12 (0.22) | 0.1                   | 0.031(6)                                   | 0.563 (6) |  |  |
|                  | TC      | 0.66 (1.14)                | 0.17 (0.21) | 0.1                   | 0.187(5)                                   | 1 (6)     |  |  |

Table 4. The emission potentials of isoprene and monoterpenoid and, sesquiterpenoid sums as the mean (standard deviation) across the trees measured per species in Helsinki in July 2022, as  $\mu g \ g(DW)^{-1} \ h^{-1}$ , at T 30°C and PAR 1000  $\mu$ mol m<sup>-2</sup> s<sup>-1</sup>. In the Wilcoxon test, the measured values were compared against the reference value (mean value across the values listed in Table S4); significant deviations (p < 0.05) are highlighted in bold. AP, *Acer platanoides*; BP, *Betula pendula*; QR, *Quercus robur*; TE, *Tilia x europaea*; UG, *Ulmus glabra*. Nobelobs, number of observations (trees) in the Wilcoxon test.

| compound         | Species | Measured<br>mean<br>BVOC<br>emission<br>potential | Reference<br>BVOC<br>emission<br>potential | Wilcoxon<br>test p-<br>values<br>(htrees llobs) |
|------------------|---------|---------------------------------------------------|--------------------------------------------|-------------------------------------------------|
| Isoprene         | AP      | 0.01 (0.01)                                       | 0.07                                       | 0.063 (6)                                       |
|                  | BP      | 0.09 (0.16)                                       | 0.05                                       | 1 (6)                                           |
|                  | QR      | 78.0 (29.8)                                       | 64.6                                       | 0.625 (5)                                       |
|                  | TE      | 0.04 (0.04)                                       | 2.75                                       | 0.031 (6)                                       |
|                  | UG      | 0.01 (0.01)                                       | 0.11                                       | 0.031 (6)                                       |
| monoterpenoids   | AP      | 0.05 (0.04)                                       | 1.72                                       | 0.031 (6)                                       |
|                  | BP      | 4.46 (6.30)                                       | 2.37                                       | 1 (6)                                           |
|                  | QR      | 0.22 (0.31)                                       | 0.90                                       | 0.031 (6)                                       |
|                  | TE      | 0.05 (0.05)                                       | 0                                          | 0.031(6)                                        |
|                  | UG      | 0.07 (0.04)                                       | 0.11                                       | 0.156 (6)                                       |
| sesquiterpenoids | AP      | 0.80 (1.06)                                       | 0.1                                        | 0.313 (6)                                       |
|                  | BP      | 0.20 (0.18)                                       | 2                                          | <b>0.032</b> (6)                                |



| QR | 0.30 (0.18) | 0.1 | 0.063 (6) |
|----|-------------|-----|-----------|
| TE | 0.13 (0.17) | 0.1 | 0.438 (6) |
| UG | 0.28(0.32)  | 0.1 | 0.438(6)  |



# 3.3 O<sub>3</sub> and SOA formation potentials of urban tree BVOCs

The calculated OFP and SOAFP differed significantly between tree species, both in Montreal and Helsinki (Fig. 5, Table 2). OFP was highest in PC and QM in Montreal and QR in Helsinki because of their strong isoprene emission potentials (Figs. 5a–c). The SOAFP differences between species were also driven by the sesquiterpenoid and monoterpenoid emission potentials (Figs. 5d–f). Thus, TC in Montreal during period I and BP in Helsinki approached the PC, QM, and QR SOAFP because of their higher sesquiterpenoid or monoterpenoid emission potentials (Figs. 5d and 5f).

Across species, OFP and SOAFP were higher in street trees than in park trees in Montreal during period II ( $F_{1,22} = 7.74$ , p = 0.011 for OFP; and  $F_{1,22} = 8.41$ , p = 0.008 for SOAFP; Table 2). In Helsinki, we observed no similar differences (Figs. 5c and 5f, Table 2).

Figure 5. O<sub>3</sub> formation potential (OFP, a-c) and SOA formation potential (SOAFP d-e) of the isoprene, monoterpenoid, and sesquiterpenoid emissions by urban tree species in Montreal in June (period I: a and d) and August (period II: b and e) and in Helsinki in July (c and f). The dots with error bars indicate the mean and the 95% confidence intervals for the OFP or SOAFP per species (n<sub>obs</sub> = 1-3) and the colours indicate the contributions of each compound group. OFP or SOAFP were calculated from the BVOC emission potentials normalised using the median temperature for the sampling period (28°C in Montreal and 27°C in

Helsinki). For the OFP or SOAFP calculated from the BVOC emission potentials normalised for 30°C, see Fig. S8. The The-different lowercase letters indicate a significant difference between species within the city and measurement period in Montreal. Species: AP, Acer platanoides; GT, Gleditsia triacanthos; PC, Populus x canescens; QM, Quercus marcrocarpa; TC, Tilia cordata; BP, Betula pendula; QR, Quercus robur; TE, Tilia x europaea; UG, Ulmus glabra.

# 3.4 Neighbourhood-scale BVOC emission potentials, OFP and SOAFP of study species



Among our study species, the isoprene emissions of oaks (QM in Montreal and QR in Helsinki) dominated the total isoprene emission potentials in the upscaling test areas in both cities (Figs. 6a–b), despite their small proportion of the total canopy area within the upscaling test areas (Figs. 6k–l). The study species most common in the canopy area of the upscaling test area in Montreal—AP, GT, and TC—were responsible for most of the total monoterpenoid and sesquiterpenoid emission potentials (Figs. 6c–and 6e). In Helsinki, most of the monoterpenoid emissions originated from BP despite its smaller proportion of total canopy in the upscaling test area, while the sesquiterpenoid emissions primarily originated from AP and TE with large canopy areas (Figs. 6d and 6f). The total OFP closely mirrored the total isoprene emission potential per land area (Figs. 6g–h), whereas the total SOAFP in both cities was impacted more by the common species and their monoterpenoid and sesquiterpenoid emissions (Figs. 6i–j).

Figure 6. Upscaled total emission potential per land area for isoprene (a–b), monoterpenes monoterpenoids (c–d), and sesquiterpenoids (e–f), as well as for their O<sub>3</sub> formation potential (OFP, g–h) and SOA-formation potential (SOAFP, i–j) of the study species within the upscaling test area in Montreal in June and August (periods I and II: a, c, e, g and i) and Helsinki in July 2022 (b, d, f, h and j). The BVOC emission, OFP, and SOAFP intensity estimates only include the individuals from our study species, whereby their proportions of the entire canopy area within the upscaling test areas are listed for Montreal (k) and Helsinki (l). The vertical brackets show the estimated total emissions or formation potentials without correction for shading (upper limit) and a stronger correction for shading (lower limit).

#### 4 Discussion







# 4.1 Urban tree BVOC emission rates and variation in the urban landscape

In this study, we provided the BVOC emission potentials for nine urban tree species, measured in the city from mature urban trees. We sampled BVOCs from park and street trees to account for variations in urban growth conditions, finding that the BVOC emission potentials slightly differed between the two site types (park and street) depending upon the city or sampling period. In Montreal, the BVOC emission potentials were higher among street trees than park trees, particularly for isoprene and sesquiterpenoids. In Helsinki, the BVOC emission potentials were generally higher among park trees than street trees, particularly for green leaf volatiles (GLVs). However, the large intraspecific tree-to-tree variation we observed within sites suggests that either inherent tree characteristics or small-scale variations in growth environments also play a major role in the variation of tree BVOC emissions.

Multiple factors can affect the emission potential differences between parks and streets. On the one hand, higher temperatures in the street environment (Bowler et al., 2010) can cause higher BVOC emission potentials, for example, because of increased risk of heat stress (Bao et al., 2023; Niinemets, 2010; Pollastri et al., 2021). In addition, high concentrations of O<sub>3</sub> trapped in street canyons (Abhijith et al., 2017; Karttunen et al., 2020) can increase the emissions of monoterpenoids, sesquiterpenoids, and GLVs (Bao et al., 2023; Ghirardo et al., 2016; Lim et al., 2024), but inhibit those of isoprene (Bellucci et al., 2023). Moreover, mechanical damage from traffic and pedestrians may be more frequent on streets, inducing the emissions of stress-related BVOCs such as GLVs and specific terpenoids (Holopainen and Gershenzon, 2010; Panthee et al., 2022; Portillo-Estrada et al., 2015). On the other hand, higher CO<sub>2</sub> concentrations on streets (Gratani and Varone, 2007, 2014) may inhibit isoprene and monoterpenoid emissions (Bao et al., 2023; Bellucci et al., 2023), while a diminished availability of water (due to impermeable street surfaces) can reduce the production and emissions of BVOCs in the long term (Bao et al., 2023; Niinemets, 2010). Finally, factors such as de-icing salts (Cekstere et al., 2020; Helama et al., 2020) and variations in the light environment (Simon et al., 2019) can also impact differences in the BVOC emission levels between park and street trees. The higher BVOC emission potentials from street than park trees in Montreal may be explained by the generally higher

temperatures and PAR on streets (Fig. S5). We found that the isoprene emission potential differences between site types were largest during period II in August when the temperature difference between parks and streets was also greatest. In addition, we observed positive correlations between ambient temperature (and PAR) and the isoprene and terpenoid emission potentials, in particular, for *Populus x canescens* (PC) and *Tilia cordata* (TC). While O<sub>3</sub> concentrations were also generally higher on streets than in parks in Montreal, they were generally low and thus unlikely to strongly impact the emissions. Yet, *Acer platanoides* (AP) showed increased sesquiterpenoid emission potentials with higher O<sub>3</sub> concentration, suggesting a higher sensitivity to O<sub>3</sub> concentrations. Compared with Montreal, Helsinki is a smaller city with a lower population density (3032 km<sup>-2</sup> in Helsinki vs 4834 km<sup>-2</sup> in Montreal) and a cooler summer climate. Consequently, we observed no differences in the park and street ambient temperatures, light intensities and O<sub>3</sub> concentrations during the sampling in Helsinki (Fig. S5). We note that sampling in Helsinki took place in July, which is generally the warmest month, but in comparison to August, it may

not carry similar cumulative effects of heat or pollution. —To explain the generally higher BVOC emission potentials from park versus street trees in Helsinki, further research is needed which attempts to identify the emissions drivers, for example, the potential stress factors not identified in this study. That the GLV emission potentials were specifically higher among park trees suggests that mechanical damage (e.g., from park management and recreational activities) or biotic effects may play a role. Biotic effects could be more important in parks in comparison to streets due to higher vegetation density supporting larger herbivore populations or more effective dispersion of herbivores and fungal spores. Overall, our results suggest that the potential street-park differences in BVOC emission potentials are not necessarily uniform between cities, even when close in climate or size, calling for further studies in cities with differing climates, populations or planting infrastructures.

Yet, because of the slight differences between park and street trees in our study, inclusion of different types of urban site types in further BVOC sampling efforts seems essential, although disentangling the effects of the differing urban environments on BVOC emissions would require more detailed exploration of the local environmental factors affecting the emissions. Similar BVOC emission potential comparisons have been conducted across rural—urban gradients. For instance, Lahr et al. (2015) reported higher isoprene emission rates in urban and suburban sites than in rural sites for *Quercus stellata* and *Liquidambar styraciflua* in Houston, Texas, USA. By contrast, Duan et al. (2023) reported higher isoprene and monoterpene emission rates in rural versus suburban and urban sites amongst a combination of broad-leaved species and two pine species in Xiamen, China. Correspondingly, Yu (2023) observed higher isoprene emission rates for *Pinus densiflora* and higher isoprene and monoterpene emission rates for *Acer palmatum* in rural versus urban forests in South Korea. If we consider the gradient from parks to streets as a continuation of the rural—urban gradient, our results from Montreal partially correspond to the results of Lahr et al. (2015) with the higher isoprene emissions in more urbanised environments. Our results from Helsinki, on the other hand, lie closer to those of Duan et al. (2023) and Yu (2023), finding higher BVOC emission potentials in less urbanised environments.

#### 4.2 Urban tree BVOC emission rates in comparison to nonurban trees

We aimed to understand whether non-species-specific BVOC emission potential database values, or stressful or beneficial characteristics of the urban environment would cause the measured urban tree BVOC emission potentials to differ from the emission factors presented in databases often used for BVOC emission budgeting or modelling. Notably, we found generally close agreement between the measured mean BVOC emission potentials per species and the reference emission potentials, with a few significant deviations.

Among species with low BVOC emission potentials, such as AP, GT (*Gleditsia triacanthos*), TE (*Tilia x europaea*), and UG (*Ulmus glabra*), the measured isoprene or monoterpenoid emission potentials were often smaller than expected, suggesting that the emission potentials in databases likely overestimate the emission potentials of little-studied, low-emitting species. Similar but more drastic differences between the measured isoprene emission potentials and genus-level emission estimates have been reported for subtropical urban tree species in Australia (Dunn-Johnston et al., 2016). More precisely, they found

that replacing the genus-level estimates with the measured species-specific isoprene emission potentials reduced the estimated total isoprene emissions of the urban study species by 97%. In contrast to isoprene and monoterpenoids, the sesquiterpenoid emission potentials from some of our study trees among the low-emitting species greatly exceeded the reference emission potentials. On the one hand, the higher-than-expected sesquiterpenoid emissions can be affected by uncertainties in the database estimates, given that the sesquiterpenoid emissions are generally low and thus less commonly measured or reported. On the other hand, given the sensitivity of the sesquiterpenoid emissions to mechanical disturbances (Duhl et al., 2008), the occasional high sesquiterpenoid emissions that also exceed the monoterpenoid emissions may have been driven by local stress factors.

Among species with known high or moderate BVOC emission potentials, we observed larger but not significant departures from the isoprene reference emission potentials. In Montreal, PC and QM (*Quercus macrocarpa*) showed, in comparison to the reference emission potentials, generally lower isoprene emission potentials during period I and higher isoprene emission potentials during period II (among PC only street trees). In Helsinki, the QR (*Quercus robur*) isoprene emission potentials also exceeded the reference emission potentials. The highest isoprene emission potential peaks occurred on warm days with intense radiation (Figs. S6 and S7), perhaps indicating a link to the thermal- and photoprotection roles of isoprene (Peñuelas and Munné-Bosch, 2005; Pollastri et al., 2021). In addition, among PC, the tree with the highest isoprene emission potentials (Fig. 4b) also exhibited symptoms of stress with dry shoot tips in the canopy.

Despite these departures, the agreement between our species-wise mean urban tree BVOC emissions and the reference emission potentials was generally close. Thus, the BVOC emission potential estimates do not appear to serve as a large source of error in creating urban BVOC emission budgets in cities like Montreal and Helsinki (small to mid-sized cities with cool, humid climates). Greater uncertainties potentially arise from estimates of tree biomass or leaf area index (Yang et al., 2022) or from inaccuracies in tree inventories (Bao et al., 2024). However, the large tree-to-tree variation in our results suggests that there are cases where using the database values is not suitable. For example, trees experiencing high temperatures and intense radiation or trees with visible symptoms of damage or stress may have larger-than-expected BVOC emission potentials. In addition, previous research reported considerable variation in the species-wise BVOC emission potentials between cities (Bao et al., 2023; Préndez et al., 2013), highlighting the importance of further studies on BVOC emission variability among urban trees globally in cities with different climates and populations.

#### 4.3 Air quality implications of urban tree BVOC emissions







Finally, we estimated the ozone-formation potentials (OFP) and SOA-formation potentials (SOAFP) for the typical urban tree species in Montreal and Helsinki, both as leaf-level potentials and tentatively upscaled to the neighbourhood level. The leaf-level OFP was highest among species with high isoprene emission rates—that is, PC and QM in Montreal and QR in Helsinki—while other BVOCs contributed minimally. When comparing species at the neighbourhood-level, the isoprene emissions of oaks dominated OFP in both cities despite their small canopy coverage in the upscaling test areas, indicating that even small numbers of isoprene-emitting individuals can considerably impact the local BVOC OFP. This result agrees with earlier

recommendations to avoid isoprene-emitting *Quercus* and *Populus* species in cases where local O<sub>3</sub> concentrations require control (Datta et al., 2021; Manzini et al., 2023). In addition, management strategies aimed at protecting already existing high-emitting trees from isoprene-inducing stress factors, such as heat or intensive light (Czaja et al., 2020; Peñuelas and Munné-Bosch, 2005; Pollastri et al., 2021), could be explored to limit their effects on O<sub>3</sub> concentrations.

In comparison to OFP, the leaf-level SOAFP varied less between species because of the larger contributions of monoterpenes and sesquiterpenes on-to\_SOA formation. For example, the SOAPF of TC in Montreal and *Betula pendula* (BP) in Helsinki were of similar magnitudes as the SOAPF of QM and QR (isoprene emitters) due to their higher sesquiterpene (TC) or monoterpene (BP) emission potentials. The same pattern also occurred when comparing species at the neighbourhood-level: the small monoterpenoid and sesquiterpenoid emissions of species with large areas of the canopy (AP and TC or TE) contributed to SOAFP per land area as much (Montreal) or more (Helsinki) than the high isoprene emissions of oaks with small areas of the canopy. Thus, to control local SOA production, species with high isoprene, monoterpenoid, or sesquiterpenoid emission potentials should be avoided (Datta et al., 2021). However, even small monoterpenoid and sesquiterpenoid emission levels from species planted in large quantities accumulate considerable SOA contributions, whereby identifying and avoiding the local environmental conditions that can lead to high emission peaks from existing trees may be more important than species selection.

We note that our neighbourhood-scale upscaling aimed to explore the urban tree BVOC effects on O<sub>3</sub> and SOA formation when accounting for the realistic distributions of our study species. More detailed information on the impact of each BVOC on SOA or O<sub>3</sub> formation, along with the more precise quantification of urban trees, including private trees and urban forest patches, their proportion of canopy coverage and leaf area index, would be necessary to comprehensively quantify the total neighbourhood or city-scale BVOC emissions and potential air quality impacts (Bao et al., 2024; Yang et al., 2022). In addition, for more accurate site-specific SOA and O<sub>3</sub> formation potentials and air quality modelling, local atmospheric conditions (such as NO<sub>x</sub> regime) and their impacts on BVOC reactivity should be considered instead of relying on fixed MIR and FAC values.

#### **Conclusions**

In this study, we performed direct BVOC measurements of urban trees and provided the isoprene, monoterpenoid, sesquiterpenoid, and GLV emission potentials for nine urban tree species. We observed large intraspecific tree-to-tree variations and city-dependent differences in the BVOC emission potentials between park and street trees, but no large deviations in the measured species-specific mean BVOC emission potentials from the nonurban reference emission potentials. While these two observations may seem contradictory, they show that despite the large tree-to-tree variability that is partly explained by differences in site types (park or street), across the species and site types, there was no consistent bias towards much higher or lower BVOC emission potentials than expected based on BVOC emission database values. However, while the urban tree BVOC emission potentials were not higher than expected, they may still impact air quality. Based on our findings, avoiding planting isoprene-emitting species would help control O<sub>3</sub> formation. To control SOA formation, however,

it seems more important to protect urban trees from stress impacts, such as mechanical damage or high O<sub>3</sub> concentrations, which may increase their monoterpenoid and sesquiterpenoid emissions. Further studies should explore the most important

stress factors impacting urban trees and potential management practices to mitigate them.

The negative air quality effects of tree BVOC emissions need to be combined with other negative (reducing airflow in street

canyons) and positive (capturing pollutants and particulate matter, reducing dispersion of pollutants from the source) air quality

impacts of trees to form a comprehensive view of the net effects of trees and tree species (see, e.g., Maison et al., 2024). To

achieve this, accurate estimates of the urban tree BVOC emission potentials are crucial. Although our results generally support

using nonurban database values as estimates of urban tree BVOC emissions in mid-sized high-latitude cities, further direct

measurements of urban tree BVOC emissions in various cities with differing sizes and climates, including during acute stress

events such as high O<sub>3</sub> concentrations, heat waves, or prologued droughts, are needed to scope where and when the database

values can be applied. Our findings regarding the slightly differing BVOC emission potentials between park and street trees

also highlight the importance of including various urban site types in future sampling efforts of urban tree BVOC emissions.

In addition to parks and streets, private trees in residential areas and urban forest patches would be interesting inclusions, given

that they make up a large proportion of the urban forest and may be managed differently (Sousa-Silva et al., 2023).

Data availability 610




Data is publicly available at DOI:10.5281/zenodo.15379394

**Author contributions** 

KR conceptualised the study, investigated, curated the data (with TT), conducted the formal analysis, wrote the original draft,

reviewed and edited the final manuscript, and acquired funding. JA and TT contributed to the methodology and reviewed and

edited the written manuscript. JB, HH, and AP conceptualised the study, provided resources, and contributed to the writing

through reviewing and editing the manuscript. AP supervised the project and acquired funding for this study.

**Competing interests** 

The authors declare that they have no conflict of interest.

Acknowledgements

We thank Guillaume Couture and the city of Montreal, and Juha Raisio and the city of Helsinki for their collaboration and

permissions to study public trees. We also thank all of the summer interns for their valuable fieldwork assistance.

# Financial support

This work was supported by Fonds de Recherche du Québec Merit scholarship programme for foreign students (PBEEE 304921) awarded to KR, as well as NSERC Alliance and Discovery grants awarded to AP (ALLRP 571966-22; RGPIN-2018-05201).

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
