# Peer review of "Variability in BVOC emissions and air quality impacts among urban trees in Montreal and Helsinki"

_EGUsphere, 2025_

## Author Comment (AC1)

[Figure]

Figure R1: Pearson's correlations between the emissions potentials of compound groups and environmental (Tamb: ambient temperature during sampling, PAR: photosynthetically active radiation during sampling, O3: ambient ozone concentration during sampling, % imperm: degree of surface impermeability within a 10-m radius around the tree (Helsinki) or mean surface impermeability of the city block (Montreal)) and physiological (WP: shoot water potential) factors that may be sources or indicators of stress. Correlations performed separately by species, city and measurement period.

[Figure]

Figure 3 alternative version. The difference between the isoprene, monoterpenoid, and sesquiterpenoid emission potentials measured from urban trees and the reference values (mean value of the emission potentials collected from the emissions databases), separately for species in Montreal in June (period I) and August (period II) and in Helsinki in July. The filled dots indicate the mean difference across the trees ($n_{obs}$ = 4–6) and the error bars show the 95% confidence intervals (some are smaller than the dot diameter). The open triangles show the difference for each park tree and the open dots for each street tree. Species: AP, *Acer platanoides;* GT, *Gleditsia triacanthos*; PC, *Populus x canescens*; QM, *Quercus marcrocarpa*; TC, *Tilia cordata*; BP, *Betula pendula*; QR, *Quercus robur*; TE, *Tilia x europea;* UG, *Ulmus glabra*.

---

## Author Response (AR1)

**List of relevant changes**

- Reducing the emphasis on stress effects in the introduction, where discussing the potential reasons why urban tree BVOC emission potentials may differ from the emission database values
- Clarifying our motivations for the selection of the two study cities and removing any direct comparison between the cities due to the differing sampling schedules.
- Rewording the methods section on our BVOC sampling methods to avoid points of confusion
- Stating more clearly the intent of our BVOC emission upscaling exercise (intended to compare species while taking into account their presence in the landscape, not to give an accurate neighbourhood-wise estimate of the BVOC emission potential per land area)
- Adding new correlation analysis to explore connections between the biogenic volatile organic compounds and potential sources of stress in the tree environment (new Figure 3), to address the small-scale variability in the urban environment, to overcome the strict street-park dichotomy.
- Adding detail to the comparison of database values and measured BVOC emission potentials to show the differences also per site type (park and street trees), modified Figure 4.
- Adding tables related to Figures 1 and 2, and Figure 4 in the main text.
- Addressing the apparent conflict between our two main hypotheses and adding discussion on when database values may be acceptable to use as urban tree BVOC emission estimates and when they may cause unintended bias. Also, stating more clearly the generalisability of our results across cities.

**Referee #1**

**General Comments**

The research addresses the important and timely topic of biogenic volatile organic compound (BVOC) emissions from urban trees and their potential impact on air quality. The selection of two representative high-latitude cities, Montreal and Helsinki, and the novel comparison between park and street habitats, provides a valuable contribution to the field. The methodology, based on direct, in-situ measurements, is commendable and yields useful data. However, despite these strengths, the manuscript in its current form suffers from several critical flaws in experimental design, data analysis, and interpretation that undermine the reliability and generality of its conclusions. These deficiencies must be thoroughly addressed before the manuscript can be considered for publication.

Thank you for your time and effort to go through the manuscript and write the constructive feedback. The comments presented points that were indeed challenges in building the statistical testing and the storyline of the manuscript, and they are very helpful to further develop the manuscript for better usability by the readers and community. We answer each specific and technical comment below, the numbered answers being the same as those in the referee comments. The line numbers refer to the manuscript version with track changes.

**Specific Comments**

- 1. The most significant flaw is the disparity in sampling strategy between the two cities. BVOC emissions were sampled twice in Montreal (June and August 2022) but only once in Helsinki (July 2022). Given the strong seasonal dynamics of BVOC emissions, this temporal mismatch makes any direct comparison between the cities scientifically unsound. For example, the higher emissions observed in Montreal in August could be due to late-season phenology or heat stress, a dynamic that was not captured in the single mid-summer measurement in Helsinki. This makes the discussion of inter-city differences highly speculative.
- We agree that the sampling schedule between the two cities, which was caused by technical limitations, is not suitable for comparisons. Direct comparisons between the two example cities were not within our initial aims, and during the data processing, we also decided against doing any direct comparisons between the two cities. Seeing the slightly differing results between the cities, we were, however, curious to explore whether some of the differences could be connected to the variations in city size and climate. As evoked by the referee, it is possible that the differing results were also influenced by seasonality on top of the effects of the city and species selection (which also differed between the two cities), and we agree that this is a valuable point to add to the discussion on the results by city (see lines 476-477, 505-506).
- 2. While small sample sizes may have been common in early BVOC research, a sample of n=3 is statistically insufficient for a variable as notoriously heterogeneous as biogenic emissions. This limitation results in low statistical power for analyses like ANOVA and prevents generalizable conclusions. The extremely large error bars in Figures 1, 2, and 4 are a direct reflection of this high variability. The authors' own caution to interpret the results "with care" does not absolve this fundamental design weakness, which undermines the validity of the study's findings.
- The large tree-to-tree variation was an important lesson learnt through the research process and will be taken into account in any further projects. Yet, as we sampled 6 trees per species and in total 30 trees per city in the field we believe that the data is still interesting to the community. The statistical tests, while low in power, give indications of the potential trends within urban areas and directions for future research. To alleviate the small sample size per species and site type, we did pool all species in the ANOVA tests, and both site types for testing against BVOC database values, so in no case did we do tests with only three trees.
- 3. The manuscript attributes differences in emissions to the binary park vs. street classification. However, these habitats represent a complex amalgam of confounding environmental factors (e.g., temperature, light, soil moisture and compaction, pollutant concentrations, human disturbance). The study itself found higher temperatures and O3 concentrations on Montreal streets but not in Helsinki, suggesting the "street effect" is not uniform and is city-dependent. The failure to systematically disentangle these factors makes the causal link between the "street environment" and emissions weak.
- Our aim was, most importantly, to sample trees in both common urban environments in order not to bias the results by either street or by park and to gain more representative mean values per species. We reformulated the relevant parts of the abstract (lines 19-21, 24-26), introduction (lines 112-113, 123-124, 128), and discussions (lines 510-515) to better reflect this aim rather than the street-park dichotomy as needed. However, we

were of course interested in exploring whether the park and street differed in BVOC emission rates, i.e. if there was any fundamental site-type effect that would override the heterogeneity within each site type (as seen, for example, for urban tree water use and water potential in Montreal, https://doi.org/10.1016/j.ufug.2025.128690). We agree that to further understand the potential street-park along with smaller spatial scale differences, more environmental data (especially soil, temperature, light environment and pollutants) would be needed in each site to disentangle these effects. For this, a study with a single urban tree species would potentially be more fruitful. As to the potential "street effect" not being uniform across cities, we found this interesting to report as it goes partly against the common intuitive opinion that street trees must be more stressed and thus, potentially larger sources of BVOCs. The nonuniformity is also reflective of the city-to-city variation reported elsewhere (see e.g. https://doi.org/10.1016/j.envpol.2013.04.003). We think that reporting this result can encourage further research in other cities to potentially understand which city characteristics (climate, infrastructure...) may explain the BVOC emission patterns in different urban environments.

To extend our data analysis beyond the street-park dichotomy and explore more precisely the potential characteristics within each site type that could impact the tree-to-tree variations in BVOC emissions, we performed a cross-site correlation analysis with factors of local environment that can be regarded as sources of stress. We discuss the results more in the answer to referee #1 comment 4.

- 4. The authors repeatedly invoke "stress" as a key driver of BVOC emissions, particularly in street environments. Yet, the only stress metric measured was leaf water potential, which showed that most trees were not experiencing significant drought stress. There is a critical lack of direct physiological indicators for heat stress, oxidative stress, or mechanical damage. This leaves the entire discussion of stress-induced emissions in the realm of speculation.
- We agree that more measurements of physiological stress indicators, such as Fv/Fm or photosynthetic capacity, would have been interesting additions to try an explain a larger portion of the tree-to-tree variation, including the observed differences between park and street environments. Of potential sources of stress, we measured WP, which, as noted also by the referee, showed no substantial drought and ambient O3 concentrations that did vary between trees and in Montreal between street and park environments. We did not have reliable quantification of invisible mechanical stressors (such as trucks hitting the tree canopy), but we did note any visual signs, including stem wounding and dry branches. In one case, the higher emission potentials of a tree individual did not match with visual signs of stress or damage: one street PC with high emission potentials of most measured compounds both in Periods I and II did have dry shoot tips in the canopy close to the measured branch (see new Fig 4 in answer to referee #2). We added these details in the discussion of deviations from the database values (lines 527-528, 551-552).

Additionally, as mentioned in the answer to referee #1 comment 3, we performed a cross-site correlation analysis with factors of local environment that can be regarded as sources of stress (see new Fig 3). With this analysis, we intend to bring into the analysis

details on the specific growth conditions and potential sources of stress that were called upon in the referee #1 comments 3 and 4, and referee #2 comment 1.

The analysis shows that the correlations between the potential stress factors or indicators with the BVOC emission potentials seem species-specific. However, as a commonality, high sun exposure (measured as PAR) or ambient temperature often correlated with higher emission potentials of terpenoids, even though the emission potentials are standardised to a certain temperature and PAR using G97 algorithm.

Among species-specific correlations was the correlation of higher O3 concentrations with higher sesquiterpenoid emission potentials among AP in Montreal, both in Periods I and II. This is even though O3 concentrations measured in the study were generally low. Surface impermeability degree in 10-m radius around the tree (in Helsinki) or as a mean of the block (in Montreal) correlated positively with PC isoprene emission potentials that were higher on the street than park, but interestingly negatively with sesquiterpenoid and GLV emission potentials of BP in Helsinki.

BVOC emission potentials often did not correlate with WP, but higher monoterpene emission potentials among QM and AP in Montreal in Period I and higher sesquiterpene emission potentials among TE in Helsinki were related to higher (less negative) water potential. This suggests that we did not see drought-induced emissions (as there was no drought), but good water status potentially supported higher emission potentials in certain species.

We added this analysis in methods (lines 263-269) and results (378-387 and new Fig 3) and in the discussion in the context of discussing stree-park and tree-to-tree variation in BVOC emission potentials (lines 497-499, 500-502). For certain high emission potentials we do not have explanation based on the additional analyses, but as certain VOCs such as GLVs or sesquiterpenes are generally stress-induced, we find it reasonable to assume that their high peaks may relate to a stress event even without a detailed knowledge of the type and severity of the potential stress.

5. The method to upscale leaf-level measurements to the neighborhood scale is overly simplistic and introduces massive uncertainty. It relies on multiple unvalidated assumptions, including a fixed LAI, approximate canopy areas derived from Voronoi polygons, and a simplified shading correction factor. Most importantly, the calculation only includes the study species, which represent a small fraction of the total canopy area in the test sites (23% in Montreal, 36% in Helsinki). The resulting neighborhood-scale emission maps (Fig. 5) are therefore of low accuracy and potentially misleading.

We agree that the upscaling exercise is simple, and it is so because the introduction of a more complex models would not yield more accurate results due to the nature of the underlying data. The idea of the exercise was not to provide an accurate representation of the neighbourhood emissions, because as mentioned by the referee, it lacks the other tree species and importantly, any private trees, but to compare the effects of species emissions and the species density in the landscape. We thank the referee for noting that this may not have been worded clearly enough in the manuscript, and we

clarified the intent of the exercise in the methods (line 297, 299-301), results (line 455, 457) and discussion (line 567, 577) to avoid misleading the reader to interpret the values as real neighbourhood scale BVOC emission potentials. Our discussion did already contain a note on the usability of our type of upscaling exercise (lines 585-589), including the important mention of the LAI (line 578).

We feel that even with the presented limitations, the exercise still provides an interesting insight into the importance of low BVOC emissions of trees that are present in large numbers. However, the exercise is not crucial to the conclusions of the manuscript, so if it is seen as not providing additional value to the manuscript, we are willing to submit the manuscript also without it if deemed necessary.

- 6. A central conclusion is that the measured emission potentials show little deviation from database values, thus supporting the use of these databases for urban trees. This conclusion paradoxically diminishes the novelty and necessity of the present study. Furthermore, the data show significant deviations for some species. Instead of a blanket statement that "database estimates are generally usable," the authors should provide a deeper analysis of why these deviations occur.
- Adding more analysis on why certain trees deviate more from the mean than others is a good point, thank you. We added these further analyses via the findings in new Fig 3 (see details in the answer to referee #1 comment 4), showing some of the specific conditions in which BVOC emission potentials can be higher than expected based on the database values. In addition, the new Fig 4 (see details in the answer to referee #2 comment 2) illustrates more clearly the single-tree deviations from the database estimates and can be used as a basis for deeper analysis, including the noticed potential effects of dry branches in the answer to referee #1 comment 3. We do also find that the mean emission potentials generally being close to database values is a useful result for further research, modelling efforts and urban planning, so we prefer also presenting this conclusion.
- 7. The authors state that for a portion of the samples, the incoming replacement air was not scrubbed for ozone. Although a post-hoc correction was applied based on literature values, this introduces an unquantified source of uncertainty. The accuracy of this correction, without rigorous validation for this specific experimental setup, is questionable and could have systematically biased the measurements of highly reactive terpenes.
- For all samples, the incoming replacement air going into the shoot chamber was not scrubbed for ozone (to allow for any effects of ozone on the tree leaves, "non-filtered ambient air", see line 182). However, we normally scrubbed ozone from air entering the adsorbent tubes (which was a side stream of the incoming replacement air or outgoing sample air), but in a portion of these samples, we could not use scrubber before the adsorbent tube taking air from incoming replacement air. As the theory-based and previously tested corrections (Helin et al. 2020) that we applied ended up being very small, and the terpene concentrations in the incoming air samples were in any case low in comparison to the outgoing sample air, we believe that any systematic bias left after the corrections would be a very minor effect on the final emission results.

- 8. The manuscript uses fixed MIR and FAC values to calculate ozone and SOA formation potentials. However, these coefficients are highly sensitive to the ambient chemical regime, especially NOx concentrations. The high-NOx environment typical of urban atmospheres can significantly alter BVOC oxidation pathways and product yields.
- We agree that taking into account the NOx regime (as well as O3 and NOx intake by leaves, and aerosol deposition on the leaves) would be crucial for modelling the realistic effects of BVOC emissions in urban atmosphere O3 or SOA concentrations. Our intent was to compare tree species and explore potential differences between site types in their potential to contribute to O3 and SOA formation, we think that it is reasonable to assume the same NOX regime for all species and sites and thus use the fixed MIR and FAC.
  - In further studies and as MIR and FAC (or other ways to represent a single BVOC potential to contribute to O3 or SOA formation) become available for more individual BVOCs, a more detailed representation of urban tree effects on air quality should become possible. We still had to rely on a mean MIR or FAC of the compound group for many monoterpenes and sesquiterpenes, although as seen in Table S5, they can differ largely between two compounds of the same group.
- 9. The study is located in two cities with similar cool, humid continental climates. The observed patterns, such as higher street emissions in Montreal versus higher park emissions in Helsinki, are likely not transferable to cities in other climatic zones (e.g., Mediterranean, arid, or tropical) where the dominant environmental stressors are entirely different. The authors must be more explicit about these geographical and climatic limitations in their discussion and conclusions.
- Our study focused on the high-latitude cities as these remain somewhat underrepresented in studies on urban tree functioning and also BVOC emissions. We did address the generalisability of our results directly in the discussion ("the BVOC emission potential estimates do not appear to serve as a large source of error in creating urban BVOC emission budgets in cities like Montreal and Helsinki (small to mid-sized cities with cool, humid climates)", line 555-556) and in the conclusions ("further direct measurements of urban tree BVOC emissions in various cities with differing sizes and climates, including during acute stress events such as high O₃ concentrations, heat waves, or prologued droughts, are needed to scope where and when the database values can be applied", 607-610). To further make the point, we added "street-park differences in BVOC emission potentials are not necessarily uniform between cities, even when close in climate or size, calling for further studies in cities with differing climates, populations or planting infrastructures" in the discussion line 510-512, and "in cities with different climates and populations" in the discussion on line 562.
- The manuscript's narrative vacillates between two somewhat contradictory messages:
  that urban environments have complex, city-specific effects on BVOC emissions, and
  that existing non-urban databases are generally adequate for urban trees. The authors must clarify what their single most important and robust scientific finding is and rebuild the manuscript's narrative to unequivocally support it.
- Our conclusion was indeed that 1) there seems to be large tree-to-tree variation in BVOC emissions by urban trees, a part of which seems to be related to the site conditions (street or park), but that 2) across all trees per species, the mean does not

importantly deviate from database values. This means that, in our results overall, there is no consistent bias towards much higher or lower emissions despite the large tree-to-tree variability. Although at first read they may seem in opposition, those are not, in fact, contradictory or mutually excluding, and are both important. However, to avoid confusion and to relate the two conclusions to each other, we now word these conclusions more clearly in the discussion and conclusions sections – thank you for pointing out the apparent inconsistency. We now more explicitly note why, in some cases, we think using non-urban databases is generally fine in cities similar to our study cities (lines 553-557), and why, in other cases, it may be more important to note the variability within urban environment (lines 557-560) (see also our previous points in conclusions, lines 606-610). We also point out the apparent contradiction between the two conclusions in the conclusions section (lines 595-597).

**Technical Corrections**

- 1. In the graphical abstract, the '2.5' in PM2.5 should be a subscript.
- We assume the referee means O3, where the 3 was accidentally superscript instead of subscript, and we will correct this, thank you for pointing out the typo.
- 2. The text relies heavily on species abbreviations (QM, PC, AP, etc.). A list or table of abbreviations should be provided at the beginning of the manuscript to aid readability.
- This sounds very helpful; we added a list of most used abbreviations (lines 36-50)

Citation: https://doi.org/10.5194/egusphere-2025-2500-RC1

**Referee #2**

**General**

The study by Rissanen et al. investigated the BVOC emission potentials of urban trees (growing in parks vs. streets) and compared them with estimates of BVOC potentials of non-urban trees from emission databases. The overall topic fits well within the scope of ACP and is highly relevant, as there is still a lack of knowledge regarding the impact of urban environmental conditions on BVOC fluxes. In general, the manuscript is well structured and written, and the study's aims and hypotheses are clearly defined. The literature cited is timely and reflects the current state of knowledge. However, I have some major concerns regarding the methodology used to evaluate the stress status of trees and the conclusions drawn from the study.

Thank you for your time and effort put in reviewing the manuscript and pointing out the potential sources of confusion and inconsistencies. The comments will greatly help us to make the manuscript more readable and to deepen the data analysis regarding the potential sources of stress and their effects on BVOC emissions. We answer the major and minor below, so that the answer numbers refer to the comment numbers. The line numbers refer to the manuscript version with track changes.

**Major comments**

- 1. The authors argue in the introduction, that urban trees are expected to release higher amounts of BVOCs compared to non-urban trees (L. 103 ff.), due to environmental stressors such as drought, heat, high ozone levels, mechanical damage (L. 80 ff.). However, with the exception of leaf water potentials, which did not indicate any water stress (L. 195), no physiological parameters were analyzed, which, in my opinion, would have been necessary, to evaluate the hypothesis.
- Even though we discuss the potential for higher emission potentials with "urban stress" in the introduction, our study was not designed to test this as a formal hypothesis (accordingly we did not formulate such hypothesis), but rather to first test whether the urban tree BVOC emissions are in line with database values or not (regardless of reason for higher / lower emissions). We also discuss in the introduction the potential error coming from BVOC emission estimates that are not species-specific but rather at a genus level. To avoid confusion regarding the study questions, we modified the introduction to emphasise the stress discussion less in comparison to the other potential sources of deviation (lines 98-99, 104-105, 112-114). We also changed the beginning of discussion section 4.2 to reflect this (line 526-527).

Yet we agree that adding other measures of physiological stress indicators, such as Fv/Fm or photosynthetic capacity would have been interesting to deeper explore the data and something to consider in further studies. Based on this comment and the referee #1 comments 3 and 4, we did perform a correlation analysis to better pinpoint which potential sources of stress could correlate with higher emission potentials of BVOCs (see answer to Referee #1 comment 4 and new Fig 3). We added this analysis in the manuscript to expand the discussion of potential stress effects (methods lines 263-269, results lines 376-387, discussion lines 497-499, 500-502)

- 2. The conclusions drawn from the study are somewhat inconsistent: On the one hand, significant differences in BVOC emission potentials between trees growing in parks and streets were found for some species and compounds; on the other hand, it was concluded, that there was no deviation in the BVOC emission potentials between urban and non-urban trees. Clearly, the low number of replicates (n = 2–3), combined with the high intraspecific variability in BVOC emissions and the variations in BVOC potentials across the two measurement periods in Montreal, makes the results difficult to interpret. However, given the differences in street vs. park trees for some species and compounds, I wonder if the authors have considered to analyze the difference between emission potentials of BVOCs measured and the reference values obtained from databases separately for each site type, instead of taking means across street and park trees (Fig. 3).
  - Indeed, our conclusion was that 1) we see large tree-to-tree variation in BVOC emissions by urban trees, a part of which seems to be related to the site (street or park) conditions, but that 2) across all trees per species, the mean does not deviate importantly from database values. Therefore, there is no consistent bias towards much higher or lower emissions despite the large tree-to-tree variability. To avoid confusion (Reviewer 1 had the same comment) and to better relate the two conclusions to each other, we now address the seeming inconsistency and word these conclusions more clearly in the discussion and conclusions sections. We now more explicitly note why, in some cases, we think using non-urban databases is generally fine in cities similar to our study cities (lines 553-557), and why, in other cases, it may be more important to note

the variability within urban environment (lines 557-560) (see also our previous point in conclusions, lines 606-610). We also point out the apparent contradiction between the two conclusions in the conclusions section (lines 595-597).

To illustrate the differences from the database values by site type, we created a new version of earlier Figure 3 (now Figure 4). In certain cases, such as in Montreal, PC isoprene emissions in Period II, TC monoterpene emissions in Period II and TC isoprene emissions in Period I, and in Helsinki BP monoterpene emissions and UG sesquiterpenoid emissions, the deviations from the database values were different between the site types (often driven by large emissions by one tree individual). Given the number of repetitions, we are hesitant to do further statistical testing with these data without pooling the park and street, but the figure will allow discussing the effect of park vs street on the deviations from the database values (and potential cases where database values are not useful), along with the impacts of potential sources of stress that could play into the larger emission potentials of certain individual trees.

We now added this detail in the new Figure 4, results (lines 400-401, 403, 405, 408)

- 3. The authors make extensive use of the supplement (21 pages!), with the results of the statistical analysis mainly presented there instead of in the main manuscript. I would suggest including the results of the statistical analysis in the main document wherever possible (e.g. Figures 1, 2 and 3). In my opinion, this would improve the article's overall readability and strengthen the results section.
- Thank you for the suggestion; we moved the tables related directly to Figs 1, 2 and 3 into the main text, see Table 2, Table 3 and Table 4. We reduced details (that are now visible in the new Figure 4) in the Tables 3 and 4 in order not to overwhelm the main text.

**Minor comments**

- L. 115 ff: Please provide a brief explanation of why these two cities were chosen. Additionally, the introduction/conclusion should state that 'common urban tree species of the northern temperate zone' were analyzed, since the chosen tree species are not representative of urban trees worldwide.
- 1. We added in the methods that high-latitude cities in the northern temperate/boreal zones are a type of city where little research has been done on the study questions, and the two cities were chosen as examples of mid-sized cities representative of this city type (lines 136-138). The two cities being on two separate continents also gave the study some geographical width and a larger selection of tree species used in urban areas. We also precised that the species selected are common within the climate zone, as suggested (line 145-146).
- L. 153: Was the flow rate of 0.08 L/min enough to transport transpired water out of the plant enclosures? In figure S2 it seems, that water condensed inside the bags, which may have impacted concentrations of oxygenated VOCs.
- The flow rate through the shoot enclosure was always 2 L/min, providing a change of the enclosure air approximately every 3 minutes. The flow rate of 0.08 L/min was used in the side streams of incoming and outgoing air that were sampled into the adsorbent tubes.

To avoid confusion in this regard, we clarify the methods section about the flow rates (see lines 177-179)

- L. 211: How stable was the incoming air? I would expect BVOC concentrations in urban environments to be highly variable, so only sampling the background once a day carries the risk of subtracting the wrong background.
- The incoming ambient air was sampled during every measurement (we drew adsorbent tube samples as a side stream of both the incoming and outgoing air always). Thus, the local ambient background per tree was always subtracted in calculating the emission rate (concentration in "out" sample concentration in "in" sample). The compound-wise variability in the incoming ambient air sample can be approximated from the Table S3, where the sapling system detection limit was calculated as the "in" sample mean concentration + 3\*standard deviation.

The daily empty bag sampling was done especially to control for any impurities within the sampling system. This build-up of impurities within tubing, pumps or filters is a slower process than the variation in ambient background between sites, so we regarded once a day sufficient.

We clarified the methods lines 176, 179, 227 to avoid confusion in this regard.

- Methods S2, please revise the sentence: "The mean (SD) concentration for α-pinene without a scrubber..." There a several values given for "α-pinene without scrubber".
- Thank you for pointing out this inconsistency, we corrected this and the following sentence.

Citation: https://doi.org/10.5194/egusphere-2025-2500-RC2

---

## Author Response (AR2)

**Responses to referee comments**

**"Dear authors,**

The manuscript is almost ready for publication. Please address the minor points that have been raised by the 2nd round of review:"

AU: We thank the editor and two reviewers for their time and effort, and for bringing up points that further help improve the manuscript's robustness and readability. We answer to each point below (starting AU). The line numbers refer to line numbers in the manuscript version with track changes.

**Reviewer 1:**

**"General Comments:**

The authors have done an exceptional job of revising the manuscript. The paper is very close to being ready for publication. I recommend it be accepted after the following minor points are addressed to further enhance clarity and consistency.

**Specific Comments:**

1. While the use of fixed MIR and FAC values is acceptable for a comparative study, the discussion in Section 4.3 could be slightly strengthened. Please consider adding a single sentence acknowledging that for more accurate, site-specific air quality modeling, local atmospheric conditions (e.g., the NOx regime) would need to be considered, as they can influence these reactivity coefficients. This adds a useful caveat for future modeling work based on these findings.

AU: Thank you for this suggestion. We agree on its usefulness to avoid the use of our results where they are not suitable. We added a corresponding phrase in section 4.3, see lines 583-585.

2. The discussion regarding the higher GLV emissions in Helsinki park trees correctly suggests that "mechanical damage or biotic effects may play a role". To add more depth, I suggest briefly elaborating on what these factors might be in a specific urban context. This could include, for example, park maintenance activities (e.g., mowing, pruning), higher recreational use leading to minor physical damage, or different patterns of herbivory in parks versus streets. A short expansion here would provide a more complete interpretation of this interesting result.

AU: As suggested, we added more potential explanations for the GLV emission potential differences between park and street trees in Helsinki, see lines 502-504.

3. The manuscript consistently uses emission potentials normalized to a standard temperature of 30°C for most analyses. However, the main figure for air quality potential (Figure 5) presents data calculated from emission potentials normalized to the median temperature of the sampling period. While the 30°C version is available in the supplement, this inconsistency in the main text may be confusing for readers trying to directly compare the emission potentials with the resulting OFP/SOAFP. For better consistency and clarity, I recommend using the 30°C normalized data for the main Figure 5, and moving the median-temperature version to the supplement if desired."

AU: Thank you for pointing out the potential confusion, we switched the figures between main text and supporting materials accordingly. See lines 438, 442-444.

**Reviewer 2:**

"I have reviewed the revised version of the manuscript entitled "Variability in BVOC emissions and air quality impacts among urban trees in Montreal and Helsinki". I appreciate the authors' thorough and thoughtful responses to all comments raised in the first round of review.

Integrating the statistical analysis from the supplement into the main text has significantly strengthened the results section. The revisions have also resolved ambiguities in the conclusions, and I believe that the manuscript now makes a coherent and well-supported contribution to current research on BVOCs and atmospheric chemistry in urban environments.

I have only two minor suggestions for further refinement:

- Line 240: Tin my opinion, the explanation of 'abnormally high values' lacks a formal definition. An outlier analysis would have been useful at this point.

AU: We added a Bonferroni outlier test to identify influential points in the linear regressions against light and temperature correction (Guenther, 1997), see lines 242-244.

- I recommend a final check for typographical errors, as there are quite a few in the current version (without track changes) of the manuscript, e.g.

L21: remove space after "database"

L26: remove the second period at the end of the sentence

L51: Check parenthesis

L100: Add space after emissions

L130: Is there a word missing before "city type"? Maybe an "a"?"

AU: Thank you for pointing out these typographical errors. We have now carefully checked the final version of the manuscript to clean any errors (including the ones highlighted here).